# Modeling Performance of Butterfly Valves Using Machine Learning Methods

**Alex Ekster \*, Vasiliy Alchakov, Ivan Meleshin and Alexandr Larionenko**

Ekster and Associates, Fremont, CA 94539, USA; alchakov@srtcontrol.com (V.A.); ivansmel@gmail.com (I.M.); larik97@gmail.com (A.L.)

\* Correspondence: alex.ekster@srtcontrol.com

**Abstract:** Control of airflow of activated sludge systems has significant challenges due to the nonlinearity of the control element (butterfly valve). To overcome this challenge, some valve manufacturers developed valves with linear characteristics. However, these valves are 10–100 times more expensive than butterfly valves. By developing models for butterfly valves installed characteristics and utilizing these models for real-time airflow control, the authors of this paper aimed to achieve the same accuracy of control using butterfly valves as achieved using valves with linear characteristics. Several approaches were tested to model the installed valve's characteristics, such as a formal mathematical model utilizing Simscape/Matlab software, a semi-empirical model, and several machine learning methods (MLM), including regression, support vector machine, Gaussian process, decision tree, and deep learning. Several versions of the airflow-valve position models were developed using each machine learning method listed above. The one with the smallest forecast error was selected for field testing at the $55.5 \times 10^3$ m$^3$/day (12 MGD) City of Chico activated sludge system. Field testing of the formal mathematical model, semi-empirical model, and the regularized gradient boosting machine model (the best among MLMs) showed that the regularized gradient boosting machine model (RGBMM) provided the best accuracy. The use of the RGBMMs in airflow control loops since 2019 at the City of Chico wastewater treatment plant showed that these models are robust and accurate (2.9% median error).

**Keywords:** airflow control; aeration system; activated sludge; machine learning; model predictive control

## 1. Introduction

Precise airflow control is necessary for many industries: HVAC, chemical, petrochemical, etc. Accurate airflow control is critical for the wastewater treatment industry because the quality of the treatment depends on dissolved oxygen (DO) concentration, i.e., it depends on the amount of air pumped into an aeration tank.

An automatic airflow control system consists of a controller, actuators, and airflow control valve. A final control element (i.e., airflow control valve) is a critically important control system component.

The most popular valves used for airflow control are butterfly valves. They are inexpensive and reliable. However, as shown in Figure 1, the installed characteristic of a butterfly valve, controlling airflow to the activated sludge system, is nonlinear.

The following phenomena explain the nonlinear relationship:

- The relationship between a disk (butterfly) position and flow is not linear;
- Static pressure in the aeration tank is a level of a magnitude higher than dynamic pressure;
- Pressure drops across the valve are low compared with the pressure loss downstream of the valves.

Traditionally, airflow is controlled using a linear proportional-integral-derivative (PID) algorithm. The control of valves with nonlinear characteristics using the PID algorithm

often causes inaccurate and oscillatory behavior of the control loops. The fact that each aeration valve is controlled individually also contributes to the oscillatory behavior of the aeration control.

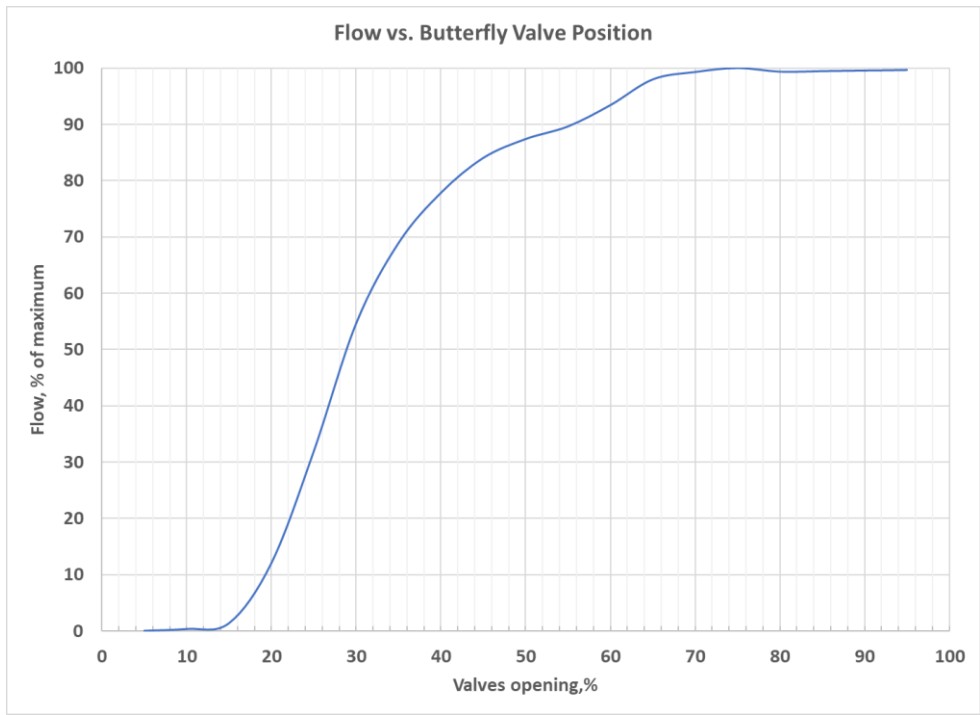

**Figure 1.** Typical installed characteristic of a butterfly valve used to control airflow to the activated sludge system.

To overcome this problem, some manufacturers developed new types of valves. For example, the iris valve (Emile Egger & Cie SA, Cressier, Switzerland) resembles the human eye in the sense that the central aperture becomes smaller and smaller as the valve closes. The Binder Vacomass Jet Control Valve (Binder Group AG, Ulm, Germany) is a venturi-style valve, and air flows through the annular space. The venturi design results in the recovery of a portion of the pressure loss, which accounts for much of its energy efficiency.

Both valves proved to be reliable and energy-efficient control elements with almost linear installed characteristics [1]. However, these valves are 10–100 times more expensive than standard butterfly valves. Because most aeration systems consist of dozens of airflow control valves, purchasing linear control valves becomes cost-prohibitive and difficult to justify.

This paper compares various modeling methods of butterfly valves performance and describes the long-term full-scale results of using two best models as a part of model predictive control (MPC) that replaced PID. There are two main contributions of this paper. First, the authors proved that using butterfly valves, it is possible to achieve practically the same accuracy of flow control as with significantly more expensive linear valves. In addition, the full-scale multi-year experience described in this paper showed that the machine-learning-based model predictive control is robust and accurate.

## 2. Background

A butterfly valve is a type of quarter-turn valve. A quarter-turn valve can open or close whenever the handle is turned 90 degrees (a quarter of a turn). The primary function of these valves is to control the flow of liquids or gas through a section of pipe. Butterfly valves are highly durable and need minimal maintenance.

Butterfly valves are made of many components. The most important one is the metal disc, commonly referred to as the butterfly. The butterfly is mounted on a rod, and when

the valve is closed, it blocks the passage of fluid or gas. The metal disc or butterfly moves a quarter turn when the valve is fully open. The passageway is unrestricted, allowing liquids or gas to pass. If the valve is opened partially, the disc will not be rotated a complete one-quarter turn; thus, it cannot provide unrestricted passage. The restriction reduces the flow through the valve.

As it was shown in Figure 1, the relationship between the disc position and flow is nonlinear. To address valves' non-linearity, many researchers [2–8] proposed to use the first principle approach for valve modeling. Tang et al. [9] used a semi-empirical approach to model pneumatic valves' characteristics. Jeon et al. [10] and Del Toro [11] successfully used computational fluid dynamics for valve modeling. Recently, machine learning started to be utilized for valve modeling. For example, Obonrkale et al. [12] used linear regression to predict cavitation in the valves. Balu et al. [13] used deep learning for valve analysis. Khalid et al. [14] used a support vector machine to detect valve stiction.

## 3. Materials and Methods

To overcome the non-linearity of the butterfly valves, the authors decided to model the installed characteristics of these valves using several methods discussed in the background section. Below is a list of the modeling methods that the authors tested:

- Formal mathematical modeling that is based on classical differential equations describing the performance of each element of the aeration systems (i.e., a first principle approach);
- Semi-empirical modeling of each control valve;
- Machine-learning-based modeling.

Each model was field-tested at the City of Chico wastewater treatment plant. The $55.5 \times 10^3$ m³/day (12 MGD) City of Chico activated sludge system consists of three aeration tanks designed as a two-pass system. There are five aeration zones in each tank. Each zone has an individual supply of air through a 100 mm diameter motorized butterfly valve, as shown in Figure 2.

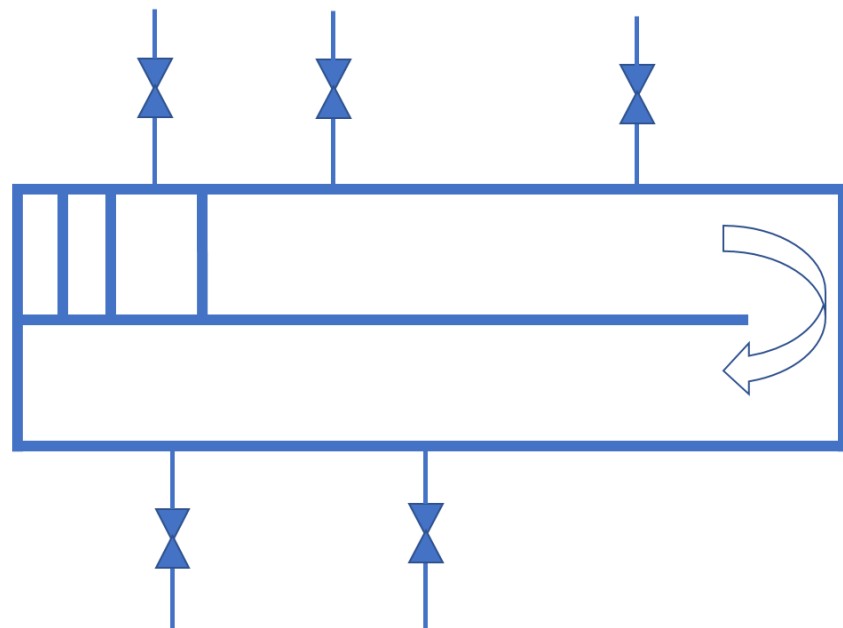

**Figure 2.** Schematic of an aeration tank (typ).

Below is a short description of each modeling approach.

### 3.1. The First-Principle-Based Modeling Approach

This approach is based on differential equations for laminar flow through the valve. The 2019 release of Simscape software [15] (Mathworks, Natick, MA, USA) was used to

model each element of the aeration system (valves, pipes, diffusers, etc.) using actual physical dimensions of these elements. Simscape software allows simulating the operation of the aeration system by solving dozens of differential equations simultaneously. A model of each component of the Chico aeration system was based on manufacturers' characteristics of this element.

This approach requires knowledge of the system's initial condition in many points of the system, including pressure drop across the valves.

### 3.2. Semi-Empirical Model

This modeling approach is based on the following semi-empirical equation describing an operation of the butterfly valve [16].

$$m = \frac{mmax}{\left(1 + a \cdot e^{-b(valveopen-c)}\right)^{1/a}}, \tag{1}$$

where *m* is the delivered flow, *mmax* is the valve nominal flow at 100% opening, and *valveopen* is the valve opening percentage.

Parameters *a*, *b*, and *c* together determine the shape of the curve, and their values were estimated based on training data. Parameter *b* determines the growth rate, *c* corresponds to the opening where the maximum growth rate occurs, and *a* characterizes the asymmetrical pattern.

Authors have developed an experimental routine of changing valves positions and measuring airflow corresponding to each position to determine coefficients in Equation (1). This routine was computer-coded and implemented for each of the 15 valves at the Chico wastewater treatment plant. Based on the experimental results, a customized airflow-valve position relationship was developed for each valve. An example of a performance surface is shown in Figure 3.

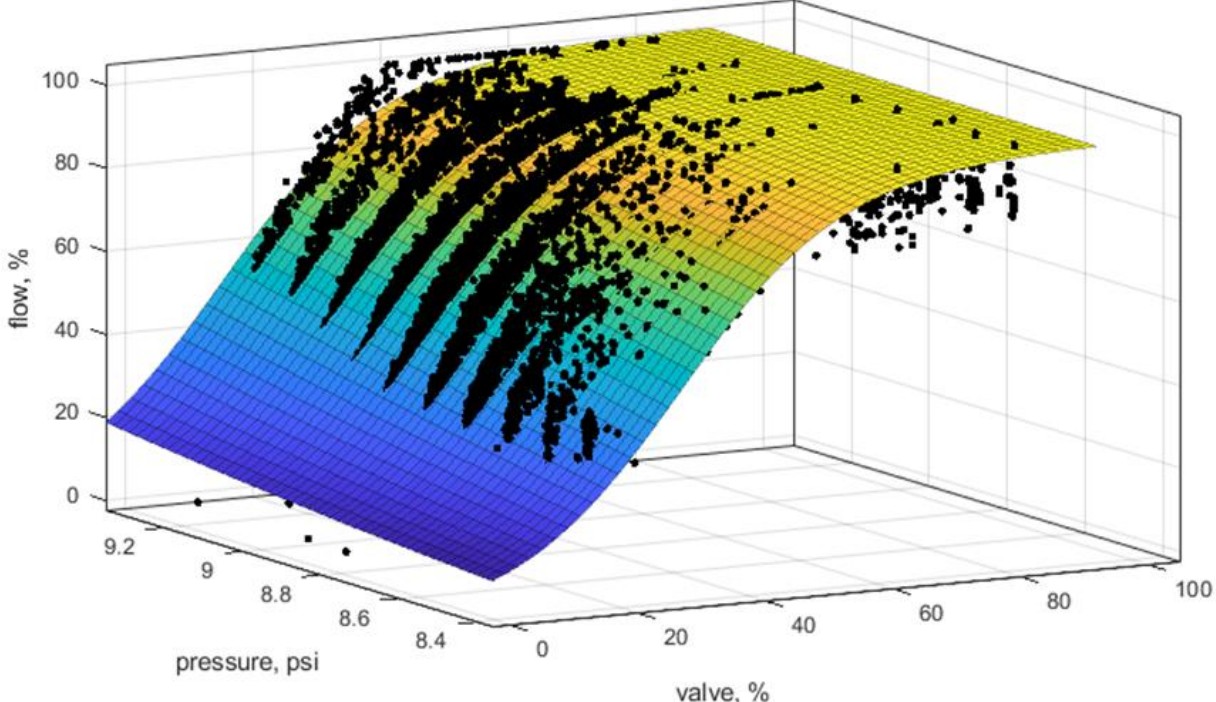

**Figure 3.** An example of a performance surface generated by a semi-empirical model.

### 3.3. Machine Learning Methods

### 3.3.1. Linear Regression

A generalized linear regression model can be described by the following equation:

$$y_i = \beta_0 + \sum b_k f_k\left(X_{i_1},\ X_{i_2}, \ldots, X_{i_p}\right) + \varepsilon_i,\ i = 1, 2, \ldots, n, \tag{2}$$

where $f(\cdot)$ is a scalar-valued function of the independent variables, $\varepsilon_i$ is the *i*th noise term, that is, the random error.

The functions $f(X)$ in Equation (2) might be in any form, including nonlinear functions or polynomials. The linearity, in the linear regression models, refers to the linearity of the coefficients $b_k$. That is, the response variable $y_k$ is a linear function of the coefficients $b$.

Traditionally, the coefficients are estimated to minimize the mean squared difference between the prediction vector $\acute{y}$ and the actual response vector $y$ (experimental data in our case).

To avoid overfitting, stepwise regression is performed by starting with a few independent variables and adding new variables one by one, and then evaluating improvements in predictive capabilities of the model using Akaike information (AIC) and Bayesian information (BIC) criteria [17].

To reduce the effect of outliers the robust linear regression is used. Models that use standard linear regression are based on certain assumptions, such as a normal distribution of errors in the observed responses. If the distribution of errors is asymmetric or prone to outliers, model assumptions are invalidated, and parameter estimates, confidence intervals, and other computed statistics become unreliable.

Robust regression uses a method called iteratively reweighted least squares to assign a weight to each data point. This method is less sensitive to significant changes in small parts of the data. As a result, robust linear regression is less susceptible to outliers than standard linear regression.

The authors tested all three above-described versions of the linear regression method.

### 3.3.2. Decision Tree

A simple decision tree is an implementation of an if-then algorithm. The decisions in the tree are followed from the root (beginning) node down to a leaf node. The leaf node contains the numeric responses.

Fine, medium, and coarse versions of the decision tree method were tested. A coarse decision tree had a few leaves (maximum number of 4 splits), a medium decision tree was medium complexity (maximum number of 20 splits), and a fine decision tree had many leaves (up to 100 splits).

In addition, the bagged decision tree was tested. Bagging (bootstrap aggregation) is used to reduce the variance of a decision tree. Several subsets of data are automatically generated from the training dataset. The selection of records in the subsets is random. Each collection of subset data is used to train their decision trees. Averages of all the predictions from different trees are used, which is more robust than a single decision tree.

Finally, the authors tested the most advanced decision tree algorithms: a gradient boosting machine (GBM) and a regularized GBM. GBMs build an ensemble of coarse trees in sequence, with each tree learning and improving on the previous one [18]. Although coarse trees by themselves have poor predictive capabilities, they can be boosted to produce a powerful selection committee. The main idea of boosting is to add new models to the ensemble sequentially. In essence, boosting attacks the bias–variance tradeoff by starting with a weak model (e.g., a decision tree with only a few splits) and sequentially boosts its performance by continuing to build new trees, where each new tree in the sequence tries to fix up where the previous one made the biggest mistakes (i.e., each new tree in the sequence will focus on the training rows where the previous tree had the largest prediction errors). A regularized GBM is a version of GBM where the algorithm tries to avoid overfitting by generating the simplest combination of decision trees and leaves on them.

### 3.3.3. Support Vector Machine Regression

A support vector machine (SWM) constructs a hyper-plane or set of hyper-planes in a high or infinite-dimensional space, which can be used for classification, regression, or other tasks. In the $\varepsilon$-SV regression that we utilized, the goal was to find a function $f(x)$ that had the deviation from all the training data smaller than $\varepsilon$ and, at the same time, was as flat as possible. In other words, the values of the errors were irrelevant as long as they were smaller than $\varepsilon$.

A kernel function is a method used to take data as input and transform it into the required form of processing data, i.e., kernel function transforms a nonlinear decision surface to a linear equation in a higher number of dimension spaces. Basically, it returns the inner product between two points in a standard feature dimension. Often to improve SWM regression, various kernels are tested. Detailed information about SWM regression and SWM kernels can be found elsewhere [19].

The authors tested the SWM with the following kernels: linear, cubic, and several versions (coarse, medium, and fine) of Gaussian Function.

### 3.3.4. Gaussian Process Regression

A Gaussian process (Gp) can be thought of as a Gaussian distribution over functions (thinking of functions as infinitely long vectors containing the value of the function at every input) [20]. Mathematically a Gaussian process can be described as a random process, where any point $x$ is assigned a random variable $f(x)$ and where the joint distribution of a finite number of these variables $p(f(x_1), f(x_2), \ldots, f(x_N))$ is itself Gaussian:

$$p(f|X) = p(f|\mu, K), \tag{3}$$

where $f = (f(x_1), f(x_2), \ldots, f(x_N))$, $\mu = (m(x_1), m(x_2), \ldots, m(x_N))$, and $K = K(x_i, x_j)$. $m$ is the mean function, and it is common to use $m(x) = 0$ as GPs are flexible enough to model the mean arbitrarily well. $K$ is a covariance function, often called *kernel*.

Kernels in (3) encode the assumptions on the function being learned by defining the similarity of two data points combined with the assumption that similar data points should have similar target values. The authors tested the following kernels: quadratic, exponential, and square exponential. Mathematical descriptions of these kernels can be found elsewhere [21].

### 3.3.5. Deep Learning

Deep learning is the name used for stacked neural networks, i.e., networks that are composed of several layers.

The layers are made of nodes. A node is a place where computation happens, loosely patterned on a neuron in the human brain, which fires when it encounters sufficient stimuli. A node combines input from the data with a set of coefficients, or weights, thereby assigning significance to inputs with regard to the task the algorithm is trying to learn. The regression forecast represents the sum of these input-weight products.

Deep-learning networks are distinguished from the more commonplace single-hidden-layer neural networks by their depth: the number of node layers through which data must pass in a multistep process. In n deep-learning networks, each layer of nodes trains on a distinct set of features based on the previous layer's output.

Traditional neural networks only contain 2–3 hidden layers, while deep networks can have 150.

### 3.3.6. Methodology of Selection of the Best Machine Learning Methods

Initially, the aeration system control algorithms utilized semi-experimental models. Field data generated using this operation were used to develop and test machine learning models. The collected database contained the following information: airflow meters readings for each valve, valve positions, and blowers discharge pressure. Data were

collected for a period of one month with 30 s frequency. The following software packages were used for the generation of machine learning models: Modeltime software package [22], h2o software package [23], and GPy software package [24].

The collected dataset was divided using a 70:30% ratio (70% of data were used for training, 30% of data were used for testing). The models' accuracies were calculated by comparing estimated valve positions and historical records of valve positions under the same flow conditions. Based on the median accuracy of valve position estimates, the best machine learning method has been selected and field-tested.

To determine the accuracy of a flow control method that used the semi-empirical model, month-long airflow data were collected with 30 s frequency. A relative control error was calculated as an absolute difference between airflow setpoint and airflow meter readings divided over the setpoint value. A similar methodology was used to determine flow control accuracy that used machine-learning-based models.

Airflow was measured by permanently installed thermo-mass flow meters (Model 410 FTB, Kurz Instruments, Monterey, CA, USA) specifically developed to control aeration valves with nonlinear characteristics. Model 410 FTB is characterized by fast response time ($T_{60} = 0.18$ s) and good repeatability (measurement noise is 0.13%).

## 4. Results

### 4.1. Simscape Mathematical Model

The airflow measured in the field was significantly (as much as 100%) different from the model prediction. As a result, the authors did not attempt to create an airflow controller that utilized this model.

### 4.2. Semi-Empirical Model

The initial testing of the semi-empirical models at the Chico wastewater treatment plant showed reasonably good accuracy. The authors integrated these models into an airflow control algorithm and operated this algorithm to control all 15 valves for over a year. The collected data showed that the median relative airflow control error was 4.5%.

### 4.3. Machine Learning Methods

A desktop comparison of the machine learning methods is provided in Table 1.

**Table 1.** Desktop comparison of machine learning methods.

| ML Method | Method Type and Accuracy in % | | | | | | |
|---|---|---|---|---|---|---|---|
| Gaussian Process Regression | Quadratic 2.41 | Exp 2.36 | Sqr. Exp 2.55 | | | | |
| Decision Tree | Fine 2.88 | Medium 3.53 | Coarse 3.39 | Bagged 3.43 | Ensemble 3.7 | GBM 2.31 | Regularized GBM 1.8 |
| Support Vector Machine | Linear 2.94 | Cubic 2.66 | Coarse Gaussian 3.2 | Medium Gaussian 2.48 | Fine Gaussian 3.49 | | |
| Linear Regression | Stepwise 3.09 | Robust 3.5 | Linear 3.55 | Interaction 3.09 | | | |
| Deep Learning (multilayer ANN) | 2.6 | | | | | | |

As Table 1 shows, the regularized GBM method provided the best accuracy. This method was tested in the field. Regularized GBM model (RGBMM) of each valve replaced the semi-empirical model of each valve. After several months of operation, the median flow control error was calculated using the Methodology described above, and the results show that it was 2.9%.

## 5. Discussion

### 5.1. Simscape Model

The poor accuracy of the Simscape model can be explained by the fact that we have utilized the original manufacturers' characteristics of the aeration system elements. However, some elements of the aeration system (for example, aeration diffusers) change aerodynamic characteristics over time due to wear and tear and bio-fouling. In addition, the Simscape model does not consider valve hysteresis, dead band, and other issues that differentiate installed valve characteristics from the ideal ones. Field calibration of the model was an option for the model accuracy improvement. However, this time-consuming exercise would need to be repeated from time to time due to continuous changes in the aerodynamic characteristics of the system. Hiring a modeler for model recalibration is not cost-effective for most treatment plants; therefore, the authors abandoned the first principle modeling approach. Dias et al. [25] also rejected the utilization of the first principle-based modeling method for the same reason.

### 5.2. Semi-Empirical Model

Operation of the airflow control system that utilized semi-empirical valve models showed that these models are robust and reasonably accurate (median error 4.5%). While the model's recalibration requires an expert's involvement, this involvement is minimal, and therefore, the cost of model recalibration is relatively small.

### 5.3. Machine Learning

The desktop investigation of the machine methods' accuracies revealed that the GBM methods showed the best accuracies. We hypothesize that the GBM methods' flexibility allows describing the highly nonlinear nature of the butterfly valve's position–airflow relationship the most accurately. Since GBM methods do not use formal mathematical relationships, they can consider such phenomena as hysteresis, stiction, dead-band, and the effect of adjacent valves operation on the valve for which the model was developed. GBM methods have another advantage over other tested methods—the speed of model development. It takes less than a minute per valve to establish a model with relatively small errors. Even the optimization routine of tuning parameters (hyperparameters) is a relatively fast process. Surprisingly enough, deep learning methods did not perform as well as some other methods. The difficulty of tuning deep learning models using a generic pre-programmed optimization routine and the relatively small training database size could explain this unexpected result.

A regularized GBM method was selected as a method of choice for the development of control signals in the airflow control algorithm. While GBM methods are formally called supervised learning methods, the optimization routine was easy to automate, and the models' recalibration was performed automatically when the monthly median flow error exceeded 5%.

The Chico plant experience showed that the machine learning modeling method provided better accuracy than both a formal mathematical model and a semi-empirical model. While, unlike the latter methods, machine learning methods require the existence of an extensive experimental database, machine learning modeling requires significantly fewer efforts and time than other methods. The RGBMM-based MPC algorithm has been in operation since 2019. It contributed to significant improvements in the overall control of the Chico activated sludge system, leading to more than 50% energy savings. Water and

Waste Digest recognized the machine learning-based control at the Chico activated sludge system as one of the 2020′s ten best projects nationwide in the water/wastewater industry.

Based on the results of this study, it is reasonable to assume that machine learning methods, mainly regularized GBM, could successfully compete with the traditional mathematical modeling of large hydraulic and aeration distribution systems. A comparison of these two approaches could be an area of future research.

## 6. Conclusions

Accurate airflow control at the activated sludge plants could be achieved either by using expensive linear valves or by utilizing inexpensive butterfly valves in combination with model predictive control (MPC). If butterfly valves are used, it is advisable to use butterfly valve models as a part of MPC. The proposed semi-empirical modeling approach is simpler and requires less effort than a first-principle-based formal mathematical modeling. Still, it requires performing specific experiments and modeling expertise. Furthermore, model recalibration could not be completely automated. On the other hand, machine-learning-based models can be developed offline using historical data, while the models' recalibration could be completely automated on-site. Machine-learning-based models also allowed achieving the best flow control accuracy (median error 2.9%). Among the machine learning methods tested, the regularized gradient boosting machine model (RGBMM) was the most accurate and required just minutes for model development. Since 2019, RGBMMs have been used at the $55.5 \times 10^3$ m$^3$/day (12 MGD) City of Chico wastewater treatment plant. A two-year, full-scale operation showed that flow control using RGBMMs is accurate and robust. The application of machine learning methods to the modeling of large hydraulic and aeration distribution systems could be an area of future research.

**Author Contributions:** A.E., Grade V operator, was responsible for the overall project development and management, investigation of machine learning methods, design of field studies, and revision of the manuscript. V.A. was responsible for field data collection, data analysis and computer coding. I.M. developed the Simscape-based mathematical model and semi-empirical model. A.L. optimized and customized RGBMM for the City of Chico control system, participated in data analysis and computer coding. All authors have read and agreed to the published version of the manuscript.

**Funding:** Field testing of the models was a part of a control system commissioning process that the City of Chico funded. The City of Chico funded customization of the model-based control system. Project No 850-670-5400.

**Institutional Review Board Statement:** Not applicable.

**Informed Consent Statement:** Not applicable.

**Data Availability Statement:** The control system that utilizes the described models is a property of the City of Chico and so are all field data used for the models' development and models' verification. The authors do not have City permission to make these data public or the control algorithms customized for the City.

**Acknowledgments:** The authors acknowledge the City of Chico wastewater treatment plant staff for helping implement the machine-learning-based aeration control. Commissioning of machine-learning-based aeration control was funded by the City of Chico.

**Conflicts of Interest:** The authors did not participate in the development of Simscape software nor have any financial interest in Mathworks Inc. The authors did not participate in the development of machine learning methods described in these studies nor did they participate in the development of machine learning computer packages utilized in these studies. The authors are responsible for developing the commercial DO/Nmaster model predictive control software package that uses a variety of models, including RGBMM.

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
