# Peer review of "Modeling Performance of Butterfly Valves Using Machine Learning Methods"

_sustainability, doi:10.3390/su132413545_

Round 1
Reviewer 1 Report
1. Authors should discuss the flow given in Fig. 1. It is not obvious for readers what orange and blue lines represent (it should be marked on the graph).
2. Research goals and contributions should be stated more clearly in the Introduction section.
3. Background and literature review are completely missing from the paper.
4. Authors should avoid using we/ours throughout the paper.
5. Instead of "see Figure", authors should use "as shown in Figure..."
6. Figure 3 quality should be enhanced if possible, it is not sharp enough.
7. The obtained results must be discussed in more detail.
8. Authors should explain the results in Table 1.
9. Authors should explain in detail the dataset used for training of the models.
10. Discuss the limitations of the proposed method.
11. Conclusion is just a bullet list of facts. Authors should rephrase it completely. Summarize the results, mention limitations, and suggest a possible future work there.
12. Bibliography is very limited. Add more recent references. Include the following:
https://www.mdpi.com/2227-7390/9/17/2068
https://link.springer.com/chapter/10.1007/978-3-030-85577-2_33
https://ieeexplore.ieee.org/abstract/document/9072559
https://www.sciencedirect.com/science/article/abs/pii/S0378778819337879
https://www.taylorfrancis.com/chapters/edit/10.1201/9781003111290-17-21
Author Response
Please find attached the reply

Reviewer 2 Report
In this paper, the authors aim to recover the accuracy related to valves with linear characteristics while using non-linear valves. My opinion is that the paper is unclear in many passages and an adequate amount of references has not been provided. Moreover, some concepts concerning the machine learning models are quite confused. For example, it is not true that deep learning is unsupervised learning, as claimed in the paper: many supervised algorithms are included in the context of deep learning.
Author Response
Please find attached the reply

Round 2
Reviewer 1 Report
The authors addressed all comments from the previous round. Paper can be accepted in the current form.
Author Response
We would like to thank the Reviewer for time and effort.
Reviewer 2 Report
The paper improved with respect to the first version. My last request is to add more details concerning the ANN used in the experiments. In a ANN there are a lot of hyperparameters to be tuned, how did you do that? Which are the resulting values?
Author Response
Dear reviewer,
Thank you for your previous comments that helped improve the manuscript's quality.
One of the goals of this paper is to show that using preprogrammed computer packages, even a non-expert in machine learning could use machine learning methods to solve a practical problem. h2o is one of these packages. h2o, while generating models, automatically optimizes hyperparameters. h2o routine for deep learning method hyperparameters optimization is provided below
https://github.com/h2oai/h2o-3/blob/master/h2o-automl/src/main/java/ai/h2o/automl/modeling/DeepLearningStepsProvider.java.
There is little doubt that experts could develop more accurate models than the models generated by preprogrammed in h2o and other used packages model automatic generation routines. However, hiring experts may not be an option in some cases, and the project described in this manuscript is an example of these cases. So, this paper, written for practicing engineers, who are not experts in machine learning, made a case that while the automatically generated models may not be the best models they still provide improvements compared with traditional linear-based calculations. As renowned George Box said:” all models are wrong but some of them could be useful”. https://en.wikipedia.org/wiki/All_models_are_wrong.
Thank you, again
